



# Local-scale uncertainty of seasonal mean and extreme values of in-situ snow depth and snow fall measurements

Moritz Buchmann[1,3], Michael Begert[2], Stefan Brönnimann[3,4], Christoph Marty[1]

1WSL-Institute for Snow- and Avalanche Research SLF, Davos, Switzerland

2Federal Office of Meteorology and Climatology MeteoSwiss, Zürich, Switzerland

3Oeschger Centre for Climate Change Research, University of Bern, Switzerland

4Institute of Geography, University of Bern, Bern, Switzerland

*Correspondence to*: Moritz Buchmann (moritz.buchmann@slf.ch)

**Abstract.**

Measurements of snow depth and snowfall on the daily scale can vary strongly over short distances. However, it is not clear if there is a seasonal dependence in these variations and how they  impact common snow climate indicators based on mean values, as well as estimated return levels of extreme events based on maximum values.

To analyse the impacts of local-scale variations we compiled a unique set of parallel snow measurements from the Swiss Alps consisting of 30 station pairs with up to 77 years of parallel data. Station pairs are mostly located in the same villages (or

within 3 km horizontal and 150 m vertical distances).

Investigated snow climate indicators include average snow depth, maximum snow depth, sum of new snow, days with snow on the ground, days with snowfall as well as snow onset and disappearance dates, which are calculated for various seasons (December to February (DFJ), November to April (NDJFMA), and March to April (MA)). We computed relative and absolute error metrics for all these indicators at each station pair to demonstrate the potential uncertainty. We found the largest relative

inter-pair differences for all indicators in spring (MA) and the smallest in DJF. Furthermore, there is hardly any difference between DJF and NDJFMA which show median uncertainties of less than 5% for all indicators. Local-scale uncertainty ranges between less than 24% (DJF) and less than 43% (MA) for all indicators and 75% of all station pairs. Highest (lowest) percentage of station pairs with uncertainty less than 15% is observed for days with snow on the ground with 90% (average snow depth, 30%).  Median differences of snow disappearance dates are rather small (three days) and similar to the ones found

for snow onset dates (two days). An analysis of potential sunshine duration could not explain the higher uncertainties in spring. To analyse the impact of local-scale variations on the estimation of extreme events, 50-year return levels were quantified for maximum snow depth and maximum 3-day new snow sum, which are often used for prevention measures. The found return levels are within each other's 95% confidence intervals for all (but two) station pairs, revealing no striking differences.

The findings serve as an important basis for our understanding of uncertainties of commonly used snow indicators and extremal

indices. Knowledge about such uncertainties in combination with break-detection methods is the groundwork in view of any homogenization efforts regarding snow time series.





## 1 Introduction

Snow, in all its forms, is of great social and environmental importance. Implications can be found in many fields as diverse as
ecology, climatology, hydrology, tourism, and natural hazards. All measurements of the snow cover are dependent on the local
characteristics of the site. Especially, exposure to wind or solar radiation, but as well as nearby buildings or trees may have an
impact on the measured snow quantities. In an ideal world this would not matter as the basic guidelines (e.g. (World
Meteorological Organization et al., 2018) tell to measure snow in a flat, not wind-exposed measurement field, which is at least
in the same distance from a building or tree as the height of the obstacle. In reality, manual measurement locations do
sometimes not fulfil these basic guidelines for many reasons like availability of suitable terrain, of possible observers or easy
access to the site. Due to this fact, the variability of the measured snow quantities on the 1km scale (i.e. next to an open field)
may be smaller than the variability on the 10-meter scale (e.g. south or north of a house). In theory, this bias introduced by
sometimes not ideal measuring locations may be season dependent as we hypothesize that the wind impact is mostly relevant
during the accumulation season (availability of loose snow) and the solar impact during the ablation season (more available
melt energy). The knowledge about such a possible season dependent bias is important as it helps to understand existing
inhomogeneities. Furthermore, this information is invaluable in view of homogenization efforts of snow data series.

Moreover, climatological applications and studies usually focus either on the meteorological winter DJF (e.g. (Scherrer and
Appenzeller, 2006)) or on the period NDJFMA (e.g.(Marcolini et al., 2017) ). However, winter in the Alps is not simply
restricted to these three or six months. In contrast, for many ecological and hydrological applications the melting spring snow
cover is the main interest (e.g. (Brown and Robinson, 2011; Livensperger et al., 2016; Zampieri et al., 2015)) and for some
applications even the onset of the snow cover (Roland et al., 2021). We therefore analyse the variations of important and
commonly used snow climate indicators for seasonal effects. Additionally, snow onset and disappearance dates are introduced
as they are important for snow phenology, which is especially crucial for ecological purposes.

Extending the data set of parallel time series introduced by Buchmann et al. (2020) in available number of stations, months
and years, enables to investigate the impact of the above mentioned bias introduced by sometimes not ideal measuring
locations, hereafter referred as "local bias", to answer the following questions:

1) Behaviour of the "local bias" as a function of time series length (sections 4.1 and 5.1) to check whether relative
percentage deviations are dependent on the length of the parallel series.

2) Seasonal dependence of the "local bias", analysing the extent to which the various seasons contribute to variations
observed in indicators derived from parallel snow series (sections 4.2). To further explore the effects at the
beginning and end of the snow season, snow onset and disappearance dates are used to put the seasonal variations
into context and to test the hypothesis that the beginning and especially at the end of a snow season are most
sensitive to the «local bias» for snow climate indicators (Sections 4.3 and 4.4).





3)  Impact of the "local bias" on return levels based on a commonly calculated return period of extreme events as
such values and its uncertainties are often used to answer engineering questions for prevention measures (Section
        4.5).

The paper is organised as follows: Section 2 introduces the data set, Section 3 covers the statistical methods used for the analyses. Results are presented and discussed in Section 4. Conclusions are drawn in Section 5.

## 2 Data

Our data consist of daily manually measured snow depth (HS) and height of new snow (HN) maintained by two separate institutions; SLF and MeteoSwiss. To obtain parallel series, station pairs were constructed by combining stations into pairs which are located within 3 km (and 150 m vertically) of each other. The mean horizontal (vertical) distance in the data set is 1 km (50 m). Most, but not all pairs consist of one SLF and one MeteoSwiss station (see Buchmann et al. (2020) for more details). The so defined set consists of 30 (24 for HN) station pairs between 490 and 1770 m a.s.l. with complete data between
October and May (September and June in extreme cases). It includes one station pair with 77 and 10 station pairs with more than 50 years of parallel data and incorporates a total of 1338 station-years and covers the time period from 1943 to 2020. However, not all station pairs cover the same length. Table A1 shows the various available time periods for each station pair. Six station pairs are excluded from all calculations involving HN due to irregular measurement procedures in the past, manifested by clusters of cases where HN equals to HS minus HS from the previous day.

For our analyses, we focused on derived snow climate indicators as seasonal values. The main variables are: average snow depth (HSavg), maximum snow depth (HSmax), sum of new snow (HNsum), maximum 3-day new snow sum (HN3max), days with snow on the ground, defined by HS of at least 1cm (dHS1) and days with snowfall of at least 1cm (dHN1). Figure A1 gives a quantitative impression of the data set by depicting elevation and mean values for HSavg and dHS1.

Availability and quality of the corresponding metadata records (coordinates, observer) is an issue. Although we managed to
compile complete metadata records, there is no guarantee that they are always precise and correct. Theoretically, the exact locations of the snow measurements are known, however, especially in the past, only approximate coordinates are recorded. The main reason is the general unconsciousness for the importance as to where the snow measurements are actually conducted. Further, in the case of some MeteoSwiss site, the coordinates refer to the main meteorological measurements and the snow measurements may have been conducted on a slightly different spot. Also, sometimes decades have passed between station
visits thus resulting in lacking information.

Potential sunshine durations are obtained with the help of Swisstopo's digital elevation model DHM25 which has an accuracy in Alpine terrain of 5-8 metres.

The basis for all our analyses, the indicator data set, is available from Buchmann et al. (2021).



## 3 Methods

To be able to compare and quantify the differences of the various snow climate indicators, we use relative percentage differences (RPD), calculated according to eq. 1 for each indicator (i) and station pair (X-Y) with the number of years denoted by n and k indicating the actual year. These RPDs are expressed as seasonal mean values for DJF, NDJFMA, and MA or monthly mean values. A potential influence of observational length on RPD is investigated by plotting the number of available parallel years against mean RPD for each indicator and station pair in the data set.

$$RPD := \frac{1}{n}\sum_{k=1}^{n}\left|\frac{X_k^i - Y_k^i}{mean(X_k^i, Y_k^i)}\right| \tag{1}$$

Absolute differences (absD) are calculated accordingly (see equation 2).

$$absD := \frac{1}{n}\sum_{k=1}^{n}\left|X_k^i - Y_k^i\right| \tag{2}$$

We defined the current hydrological year as the period from 1 Sep of the previous year to 31 Aug of the current year.

Snow onset (Dstart) and disappearance dates (Dstop) are defined as the first (Dstart) and day after the last day (Dstop) of the longest period with continuous snow cover. For the purposes in this study, we additionally allowed gaps of up to three consecutive days with no snow cover during the season. The chosen gap length allows to include full seasons in case they were fragmented in the middle of the winter by a maximum of three days without snow. Such an approach corresponds much more to the experience of the biotic world than just using the duration of the longest continuous snow cover. This definition guarantees a continuous snow cover throughout the snow season which is important for applications involving ecology, albedo, or climatological analyses.

In order to assess the impact of the local scale differences on the long-term temporal changes of snow onset and disappearance dates Theil‐Sen linear slopes (Sen PK, 1968; Theil H, 1950) are calculated for each station pair. For each station we calculated absolute changes (AC) defined according to eq. 2 as:

$$AC := fitted\ value\ at\ end - fitted\ value\ at\ beginning \tag{3}$$

To further investigate the potential impact of the beginning and end of the snow season on local bias, median differences of daily snow depth measurements for each station pair are calculated, separately for the first (accumulation) and last (ablation) 60-day periods of the hydrological year.

$$Median\ difference := median\ HS(X) - median\ HS(Y) \tag{4}$$

To investigate a potential influence of the "local-bias" on the differences in snow disappearance dates, we compared them to the difference in potential MA sunshine duration hours (Sdur) for a selection of station pairs with good quality metadata. Sdur are obtained as daily values with the help of a digital elevation model  and geographical information system (GIS) software. This calculation depends on the accuracy of the coordinates of the measurement sites.



To analyse the impact of the "local bias" on potential extreme events based on annual maximum 3-day new snow sum HN3max (e.g. Bocchiola et al., 2008) and annual maximum snow depth HSmax (e.g. Marty and Blanchet, 2012), return levels for fixed (50-year) return periods are calculated for each station and indicator using the R-package extRemes and standard settings (GEV, estimation method MLE, and 95% confidence intervals). For three station pairs (CAV, KUB, ZER), Gumbel instead of GEV is used due to bad model fit. The analysis is based on data for NDJFMA.

## 4 Results and discussions

### 4.1 Time series length and relative percentage differences

To investigate a possible relationship of RPD with time series length we plotted the mean RPD (season NDJFMA) against the length of the underlying parallel time series for each indicator (Figure 1). Here we found no clear pattern. RPD are between 10 and 50% for all station pairs and indicators. The values are in line with findings from Buchmann et. al. (2020) for a 25-year common period. However, the variations within each indicator show some visual differences; with HSavg having the largest (20%) and dHS1 the smallest (8%) variations. Outliers are caused by the same station pairs for all indicators. There is no difference between the shorter and the longer time series. This suggests that the lengths of the time series have no effect on the RPD values. This further implies that time series of different lengths can be compared and compiled into one dataset for the purpose of this study. Moreover, this highlights the possible large differences among the various station pairs involved.

The absence of any clear relationship between observational length and RPD (Figure 1) justifies the combined use of station pairs with varying lengths. However, it could be possible that the overall variations are too large to disentangle the signal of possible breaks from the noise.





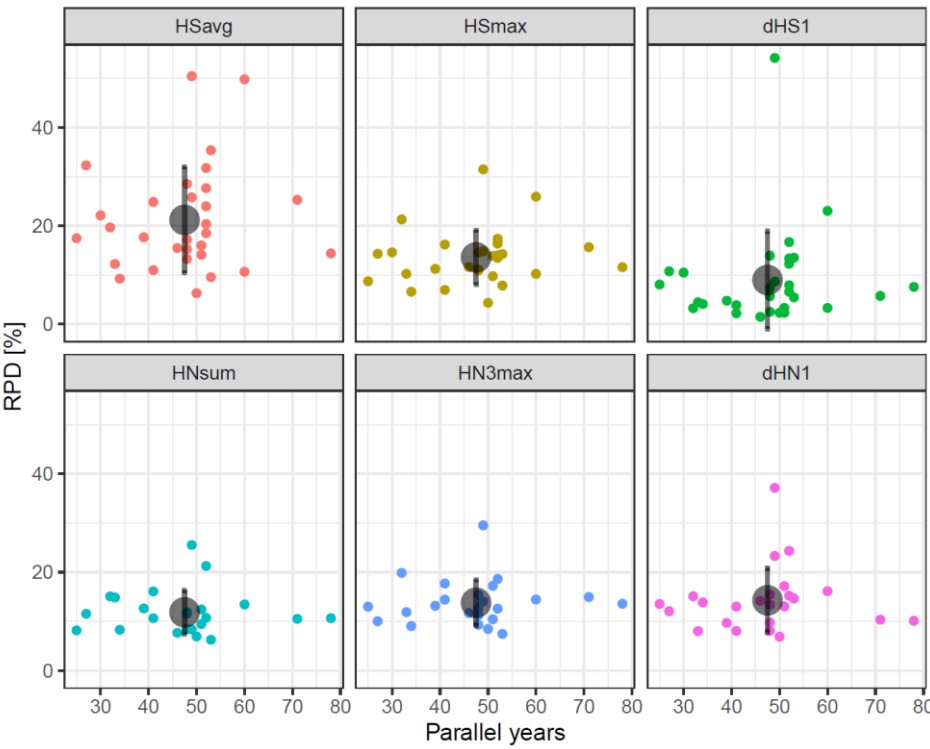

**Figure 1: Relative percentage differences (RPD) as a function of number of parallel years for six snow climate indicators. Black dots and error bars indicate mean and standard deviations.**

## 4.2 Seasonal influence

To explore the effects of various seasons on the snow climate indicators, we calculated RPD and absD for three commonly used seasons: winter (DJF), spring (MA), and the entire season November to April (NDJFMA), as well as for each individual month. Figure 2 summarizes the results. Here we found that RPD for DJF and NDJFMA are similar (difference never exceeds 5%). However, the main differences are visible in the MA period, which shows the largest RPD values for all indicators (Figure 2 plot A).

Furthermore, dHS1 has the lowest median RPD for all seasons, whereas HSavg is the one with the largest median RPD for all three seasons. Overall median RPD values for NDJFMA are between 7 and 18% and correspond to findings in Buchmann et al. (2020). Figure 2 (plot B) further reveals that largest relative variations occur early (Nov) and late (MA) in the season and the patterns look similar for all indicators.

However, the picture looks somewhat different when we look at absD (Figure 2 plots C and D). Here we see that the largest median differences are not found in the same seasons for all indicators. Figure 2 (plot D) reveals that absD for HSavg and





HSmax seem to be largest in Mar and smallest in Nov, whereas for HNsum the largest absD values are found in Jan and Feb and the smallest values in Nov and Apr. The daily HS values are generally largest in Mar and lowest in Nov thus explaining the pattern of absD for HSavg and HSmax. The same applies to HNsum; the largest monthly HN sums are observed in Jan and

165    Feb and Nov and Apr are months with usually the smallest HN sums. dHS1 shows the largest absD at first and last two months due to larger differences in thermal conditions (ground and local air temperature) in these transition months. dHN1, in contrast, shows hardly any monthly differences, because the number of snowfall events is only marginally dependent on local measurement conditions.

These monthly patterns transform to indicator dependent seasonal absD pattern (Figure 2 plot C). The different seasons reveal

170    much smaller variations compared to RPD. For NDJFMA, absD varies between 5 and 10 cm or 5 and 8 days for all indicators with the exception of HNsum. Due to the cumulating nature of the snowfall sum, the indicator HNsum always shows the largest absD values, which increases with the number of included months (NDJFMA). The same is true for the other HN count indicator dHN1, but on a much lower level. Additional information of absD allows to put the RPD into context. The lower absD values for dHS1 during DJF are no surprise, as the ground tends to be snow-covered for almost all station pairs during

175    this period, hence the difference only amounts in cases where the snow cover is quite low (couple of cm). NDJFMA shows the highest absD value for dHS1 because this period contains the beginning and end of the season.

These findings imply that the higher RPD values in the MA period for all indicators are mainly caused by smaller absolute values in this period. The only exceptions are HSavg and HSmax, which show also high absD values in the MA period. This indicates that the "local-bias" is indeed larger for snow depth during MA.

180    To further test the seasonal influence from a more general point of view, we just look at the snow depth (HS) evolution, comprising of accumulation and ablation for each station pair irrespective of the actual season. The two station pairs below 700 m a.s.l. were excluded from this analysis due to too short median winter seasons. We calculated the climatology of the daily HS differences for each station pair. We then constructed the difference series as the combination of the first (and last) 60 days for each station pair. The median of all these difference series is the thick black line in Figure 3. These two periods

185    stand for accumulation (first 60 days) and ablation (last 60 days). The 60-day period is an empirical value. Here we found that the differences observed in the first period seem to be smaller than in the last period (Figure 3). This suggests that the ablation period shows more variation than the accumulation period, which corroborates the pattern found in the absD analysis.

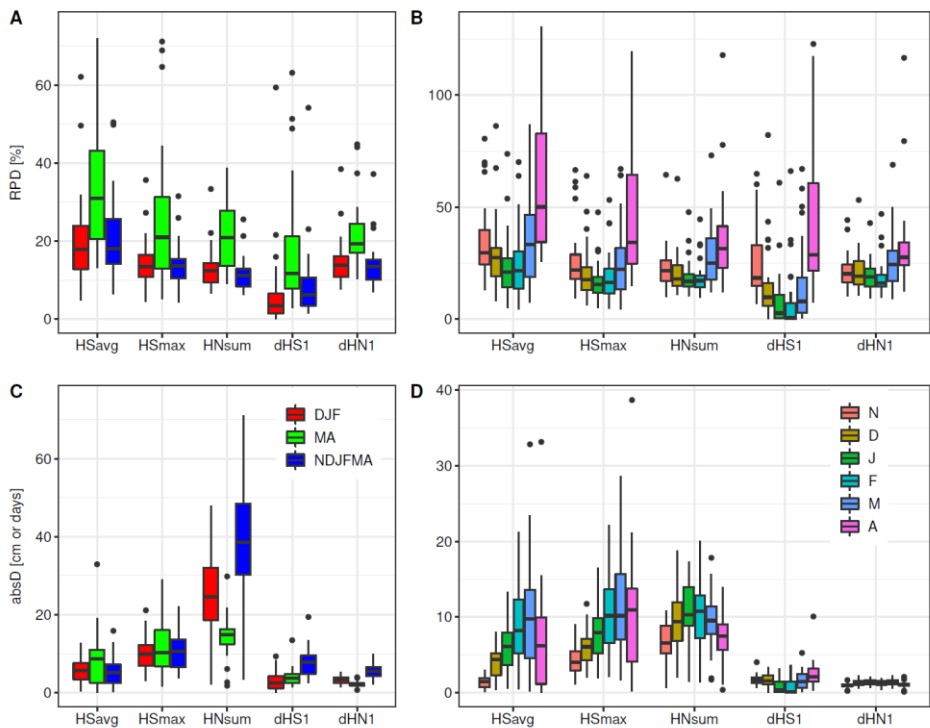

**Figure 2: Relative percentage deviations (RPD) and absolute differences (absD) for five snow climate indicators. Depicted are (left) three seasons December to February DJF (red), March-April MA (green), and November to April NDJFMA (blue) and (right) all months individually. Snow depth (HS) indicators consists of 30, snowfall (HN) indicators of 24 station pairs.**



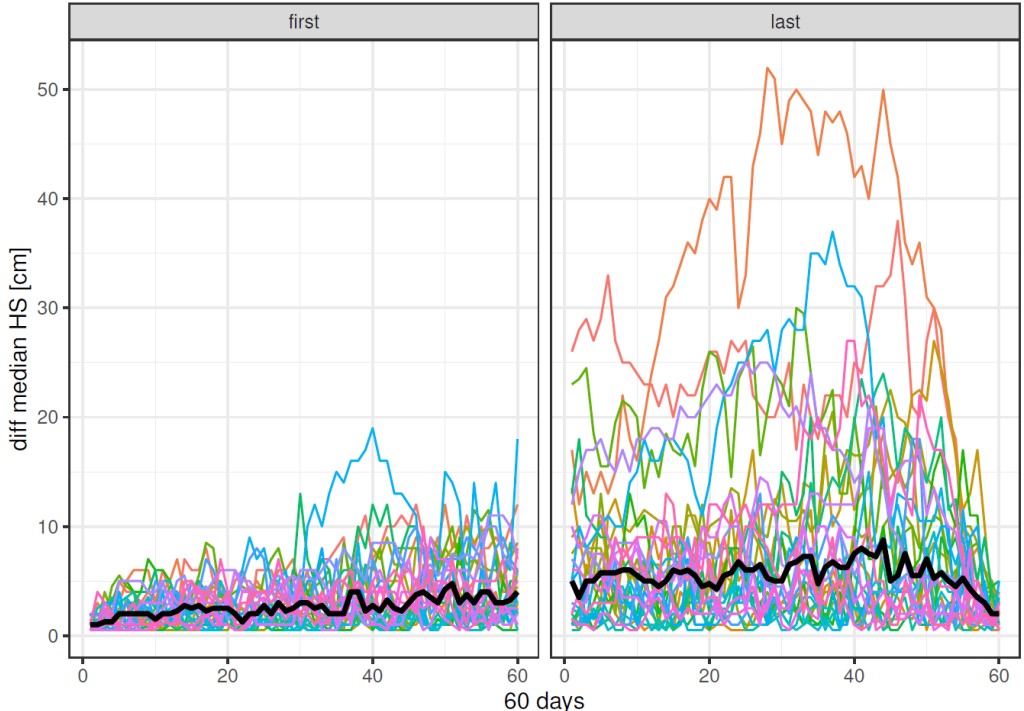

**Figure 3: Median daily absolute differences in snow depth for each station pair for the first and last 60 days of each snow season. The bold line highlights the median of all station pairs.**

## 4.3 Snow onset and disappearance dates

To further investigate the larger impact of the "local-bias" towards the end of the snow season compared to the beginning, differences between station pairs in mean snow onset (Dstart) and snow disappearance dates (Dstop) are analysed. Figure 4 (left) highlights the computed absolute differences for Dstart and Dstop. We found that for 20 (10) out of 30 station pairs differences of Dstop are larger (smaller) than differences of Dstart. In contrast to the general impression that Dstart should be the same for parallel stations, our data shows the differences of several days are not uncommon, which is probably caused by the different thermal conditions of the soil of the measurement field. Figure 4 (right) depicts the mean inter-pair differences for Dstart and Dstop. The median values over all station pairs are 2 days for snow onset and 3 days for snow disappearance dates. This corroborates the previous findings of Figure 3 that spring (ablation) shows larger variations than autumn (accumulation) with PIO being the exception. We found that for 75% of the station pairs, Dstart varies between 0 and 4 days, whereas Dstop shows slightly larger differences (0 to 6 days).

Additionally, we use absolute temporal changes of snow onset and disappearance dates, expressed as days per decade, as yet another indicator to test the uncertainty within the station pairs. For this purpose, the temporal trend in days for Dstart and Dstop are calculated for each station pair (Figure A2). As not all station pairs cover the same time periods, the pairs are aligned





from short (left) to long (right) parallel periods. Here we see that for a majority of station pairs, the direction of changes for

Dstart and Dstop is the same, Dstart tend to be associated with positive (later) and Dstop with negative (earlier) values.

Although inter-pair differences occur more pronounced and more frequently during the decline phase (spring) compared to the

accumulation period (Figure 3), the actual end of the snow season (snow disappearance date) does not show these huge

variations (Figure 4). Dstop appears to be a rather stable indicator, varying for 50% of the station pairs on average between 2

and 6 days. In contrast to what is observed in the much more complex topography on the catchment scale, this is a good result

as the relative changes derived from trend analysis of station (point) data from flat measurement fields can also be transferred

to the catchment scale, even though the absolute values may be different (Grünewald and Lehning (2015)).

Depending on the application, snow onset and disappearance dates are differently defined in spite being used frequently in

various studies focusing on snow climate  and ecology (e.g. (Foster, 1989; Kirdyanov et al., 2003; Klein et al., 2016; Peng et

al., 2013; Stone et al., 2002). Among the few stations analysed by Klein et al. (2016) are four that feature in our data set as

well. Although the time period is not exactly the same, the absolute changes in Dstart and Dstop are very similar (see Table

A2). As Klein et al. (2016) focused on single stations we can now see that the values are similar for the corresponding parallel

stations as well. This suggests that the absolute changes of Dstart and Dstop are in general quite robust.

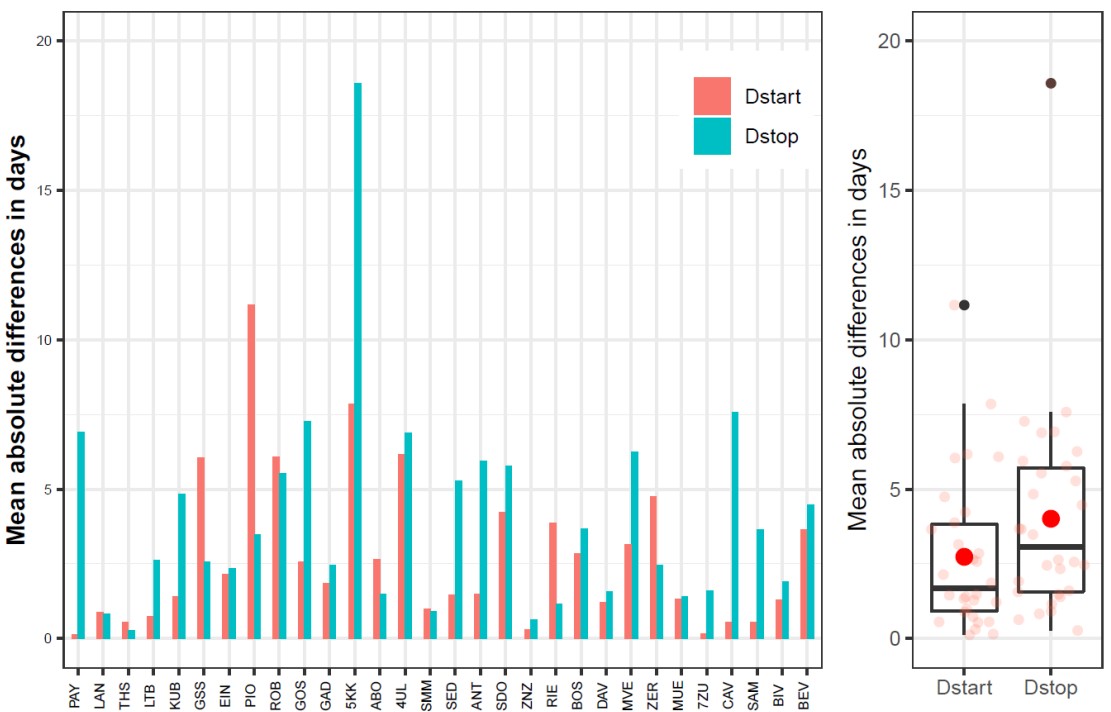

**Figure 4: Left: mean absolute differences in onset (Dstart) and disappearance (Dstop) dates for all station pairs. The station pairs**
**are ordered left to right according to their elevation. Right: Variations of mean absolute differences of Dstart and Dstop for all**
**station pairs and available years. The light red dots mark the actual values and the bold red dot indicates the mean.**



### 4.4 Influence of potential sunshine duration

To look for a possibility to explain the inter-pair differences in Dstop with available local-scale variables we compared them to differences in potential sunshine durations. Figure A3 displays the relationship of differences in Dstop and differences in potential sunshine duration during March and April for a selection of station pairs. The largest difference in a station pair of 64 hours amounts to approximately one hour per day. Mean and standard deviations of all unpaired combinations of this subset are -5 and 124 hours. Here we found no relationship between potential sunshine duration and differences in Dstop. As outlined

in section 2, our metadata is not perfect and although we limited this analysis to a selection of measurement sites with reliable coordinates, the influence of local scale obstacles cannot be detected with such an approach. As the accuracy is true for the 50 m scale, but not for 10 m, the conundrum "in front or behind a house" still remains.

We use the station pair Klosters to illustrate said issue: At a first glance, no striking difference is visible between the two stations. Elevation and surroundings are similar. One station is situated at a train station, the other next to a power station.

However, when looking closer, it is revealed that the one at the train station is located on a first-floor roof deck cutting. The second one sits just next (3 m) to the huge turbine house. Both stations are affected by their surroundings and corresponding specific exposure to wind, solar radiation and temperature is definitely different. Such differences are difficult or impossible to detect if the coordinates or metadata are not accurate. But even if these issues are known, measuring the actual impacts is virtually impractical.


### 4.5 Extreme value analyses

To test whether estimated extreme events based on annual maximum values of snow depth and snowfall differ significantly within station pairs, return levels for two indicators HSmax and HN3max are calculated for a 50-year return period using the NDJFMA season. Inter-pair differences of the 50-year return levels are small (7-8% for both indicators). We found that for all

but two station pairs (both for HN3max), the return levels of the individual stations are within the 95% confidence intervals, indicated by error bars in Figure 5. This suggests that in spite of obvious differences, in terms of return levels for 50-year return periods, the station pair values are similar or at least within each other's 95% confidence intervals.



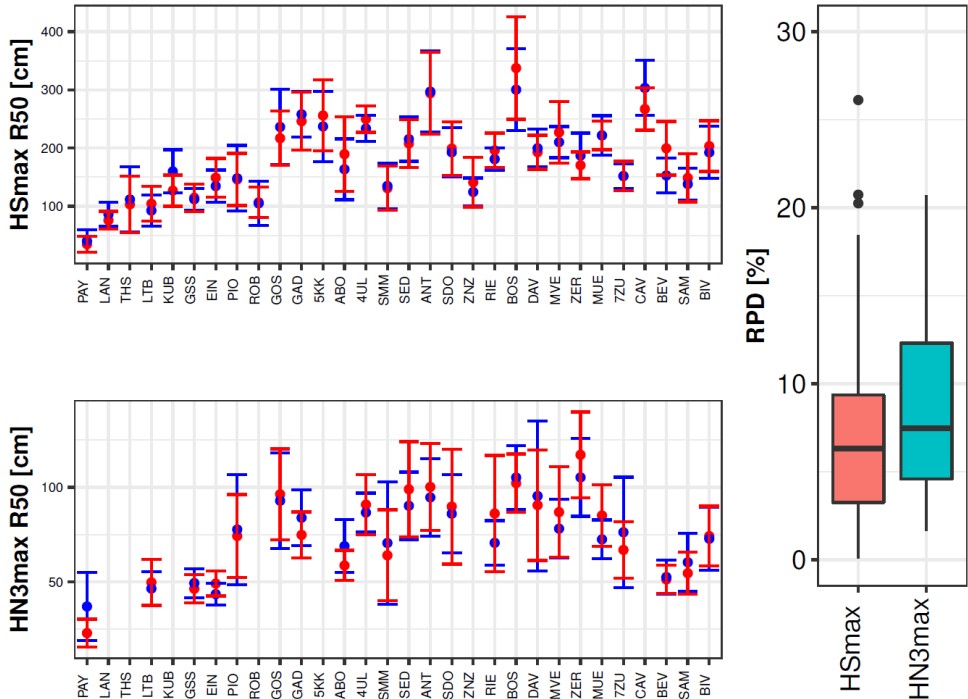

**Figure 5: Return levels for seasonal maximum snow depth (HSmax) (top left) and maximum 3-day new snow sum (HN3max) (bottom left) for 50-year return periods. Mch (slf) stations are coloured red (blue). Station pairs are ordered left to right from low to high elevation. Relative percentage differences (RPD) of return levels for HSmax and HN3max are depicted on the right.**

# 5 Conclusions

We presented the first assessment of local-scale uncertainties of common extremal indices and snow climate indicators based on a unique dataset of long-term parallel snow measurements. Analysis of estimated 50-year return periods based on annual extreme values demonstrates that the relative differences of return levels for HSmax and HN3max are less than 15% for 75% of the station pairs. For all but two station pairs these values are within each other's 95% confidence intervals, which is a good result for engineers often only depending on one single station series. On the other hand, the analysis clearly proves that just taking the return value of one station, without considering the estimation uncertainty, drastically limits the validity of the result. Regarding common snow climate indicators, the results revealed relatively small median differences for the majority of the station pairs and indicators. However, spring months (MA) have the highest relative differences (16-30%) and show more inter-pair variation (8-43%) for all indicators than DJF (2-24%) or NDJFMA (3-25%). Relative differences for these two seasons varied between 3% (dHS1) and 20% (HSavg) for all indicators, whereas HSavg also demonstrated the largest spread of all indicators. HSavg also revealed in absolute terms the highest "local-bias" (ca. 10 cm) in spring. Additionally, inter-pair median differences of the snow disappearance date (per definition a proxy for spring) displayed again more variation and





higher absolute values than the snow onset date. However, the differences between the "local-bias" of the snow disappearance
      date (2days) and the snow onset date (3days) are small; Nevertheless, there is considerable variation for some station pairs.
      As the station pairs are constructed with stations that are on average located within 1 km, the variations are most likely down
      to local-scale influences. This suggests that seasonal differences are likely caused by "local-bias", which in isolated cases can
      have huge effects and that "local-bias" is probably amplified in MA (and Nov), likely because of the larger impact of radiation
(and thus temperature). However, insufficient metadata prohibits further analyses on the influence of local-scale factors. But
      being able to quantify season dependent biases is important in itself as it increases our knowledge about existing
      inhomogeneities; especially in view of homogenization efforts of snow data series. The larger differences found in spring may
      be an indication to preferably search for inhomogeneities (breaks) in these months or to determine correction factors separately
      for the ablation season.


      Even though potential sunshine durations can vary on a local scale, especially in mountainous areas, the influence on specific
      stations is virtually impossible to determine. The influence of a tree or house in close proximity (not only regarding sunshine
      duration) on a station remains an important but unaccountable factor, especially because the exact location of the snow
      measurement stake in the past is often not known.  It is therefore not possible to attribute any local factors in the first place.
For example, a measurement field in front of a south facing wall is influenced completely differently than a station located
      behind said wall on the north facing side. These limitations inhibit the simple use of terrain indicators to explain the variations.
      Furthermore, the influence of the observer on the snow onset or disappearance date cannot go unnoticed. Officially the ground
      ought to be declared snow-free, if at least 50 % of the measurement field is snow-free. However, as simple as the instruction
      may sound, the interpretation can vary and may easily account for a difference of a couple of days.


      Generally, "local-bias" is often negligible, at least for the large majority of the stations. The problem is that problematic stations
      are not easily detectable without parallel measurements. Therefore, a larger number of neighbouring stations are needed to
      find such problematic stations and to be able to develop stable homogenizations procedures. This means that the current number
      of available long-term snow measurement sites should at least be maintained.






**Appendix**

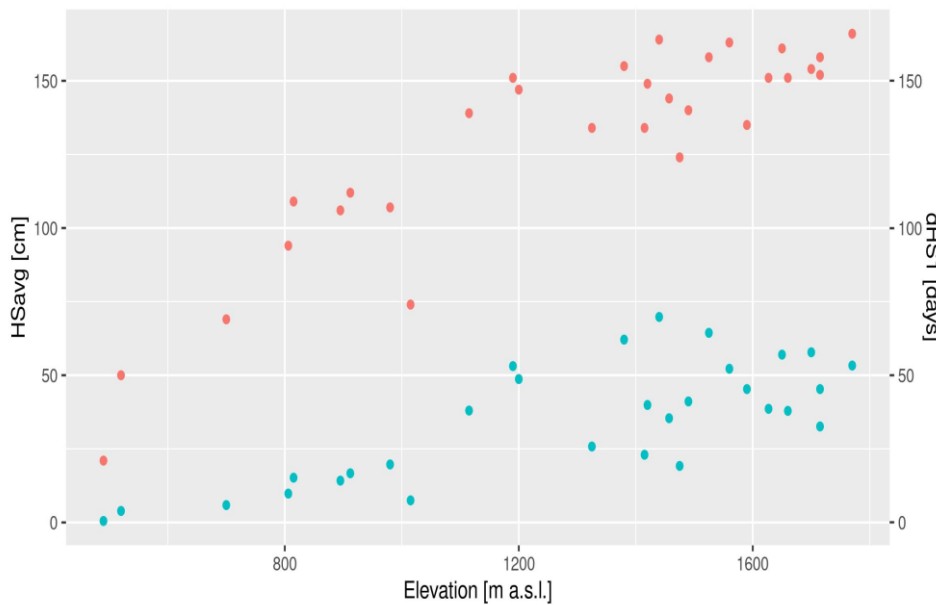

**Figure A 1: Shown are in green mean values of average snow depth (HSavg) and in red mean values of days with snow on the ground (dHS1) for each station pair and NDJFMA. Station pairs are ordered according to their elevation.**


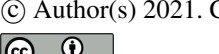

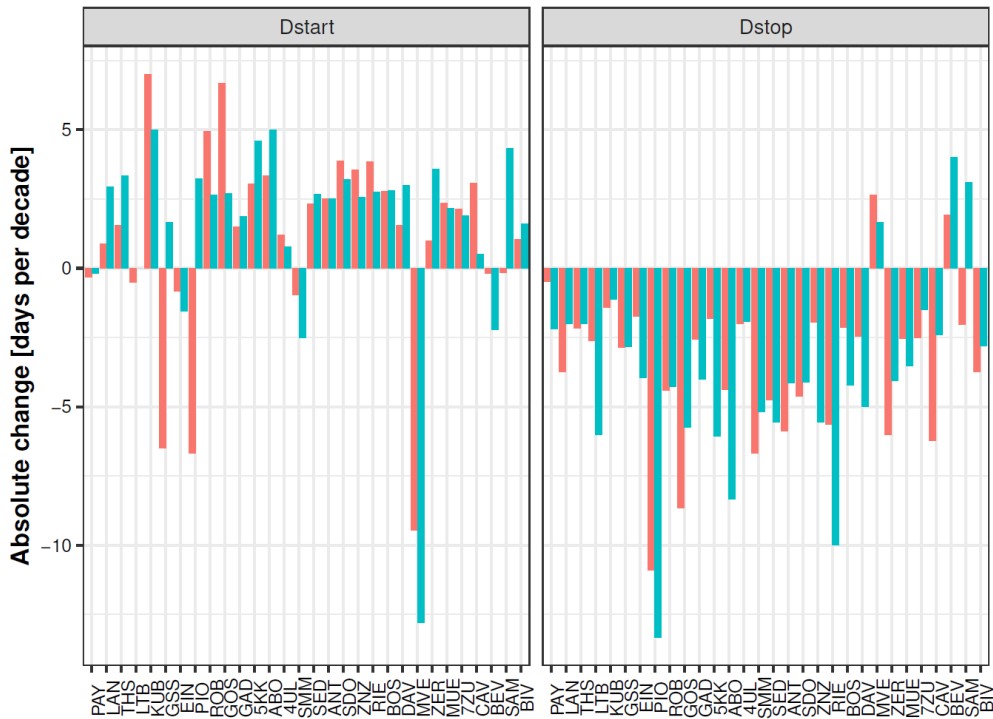

**Figure A 2: Absolute changes of snow onset (Dstart) and disappearance (Dstop) dates calculated for all available parallel periods. Station pairs are ordered according to elevation. Red and green bars indicate individual stations within a pair.**



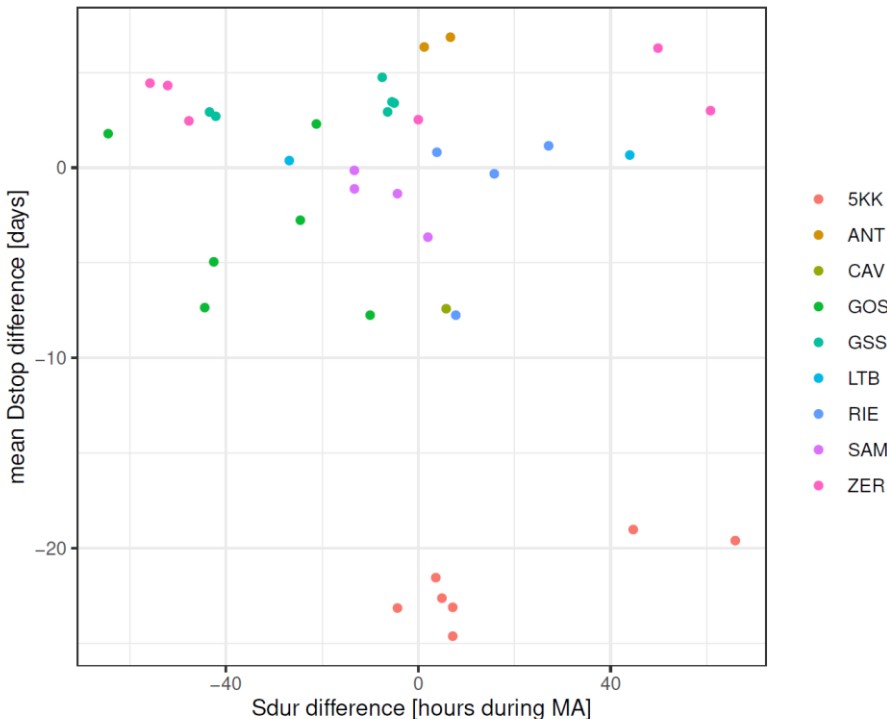

**Figure A 3: Relationship between differences in snow disappearance dates (Dstop) and differences in potential sunshine durations (Sdur) for months March and April (MA), calculated for a selection of station pairs with manually checked coordinates during different time periods.**





| mch | slf | start | stop | Elevation [m a.s.l.] | Length [years] | HN | mDstart | mDstop | mHSavg [cm] | mdHS1 [days} |
|-----|-----|-------|------|-----------|--------|----|---------|--------|--------|--------|
| PAY | PAV | 1970 | 2018 | 490 | 48 | | 15 Jan | 26 Jan | 0.5 | 21 |
| LAN | 5LQ | 1969 | 2020 | 520 | 51 | no | 03 Jan | 06 Feb | 3.9 | 50 |
| THS | 5TH | 1969 | 2020 | 700 | 51 | no | 25 Dec | 19 Feb | 5.9 | 69 |
| LTB | 1LB | 1969 | 2020 | 806 | 51 | | 20 Dec | 05 Mar | 9.8 | 94 |
| KUB | 5KU | 1991 | 2020 | 815 | 29 | no | 07 Dec | 15 Mar | 15.2 | 109 |
| GSS | EIN | 1973 | 2020 | 895 | 47 | | 18 Dec | 09 Mar | 14.2 | 106 |
| EIN | SSE | 1943 | 2020 | 912 | 77 | | 13 Dec | 12 Mar | 16.7 | 112 |
| PIO | 6AM | 1979 | 2003 | 980 | 24 | | 18 Dec | 20 Mar | 19.7 | 107 |
| ROB | 7PV | 1961 | 2020 | 1015 | 59 | no | 27 Dec | 24 Feb | 7.5 | 74 |
| GOS | 2GO | 1969 | 2020 | 1115 | 51 | | 03 Dec | 06 Apr | 38 | 139 |
| GAD | 1GA | 1969 | 2019 | 1190 | 50 | | 23 Nov | 16 Apr | 53.1 | 151 |
| 5KK | 5KR | 1968 | 2020 | 1200 | 52 | no | 22 Nov | 15 Apr | 48.7 | 147 |
| ABO | 1AD | 1966 | 2013 | 1325 | 47 | | 03 Dec | 29 Mar | 25.8 | 134 |
| 4UL | 4MS | 1950 | 2020 | 1380 | 70 | | 17 Nov | 21 Apr | 62.1 | 155 |
| SMM | 7ST | 1979 | 2012 | 1415 | 33 | | 25 Nov | 31 Mar | 23 | 134 |
| SED | 5SE | 1970 | 2020 | 1420 | 50 | | 20 Nov | 12 Apr | 39.9 | 149 |
| ANT | 2AN | 1967 | 2016 | 1440 | 49 | | 14 Nov | 01 May | 69.8 | 164 |
| SDO | 5SP | 1973 | 2020 | 1457 | 47 | | 24 Nov | 10 Apr | 35.4 | 144 |
| ZNZ | 7ZN | 1972 | 2020 | 1475 | 48 | | 01 Dec | 28 Mar | 19.2 | 124 |
| RIE | 4WI | 1975 | 2007 | 1490 | 32 | | 26 Nov | 10 Apr | 41.1 | 140 |
| BOS | 6BG | 1962 | 2014 | 1525 | 52 | | 19 Nov | 26 Apr | 64.4 | 158 |
| DAV | 5DF | 1966 | 2006 | 1560 | 40 | | 14 Nov | 27 Apr | 52.2 | 163 |
| MVE | 4MO | 1952 | 1978 | 1590 | 26 | | 06 Dec | 10 Apr | 45.3 | 135 |
| ZER | 4ZE | 1966 | 2004 | 1627 | 38 | | 17 Nov | 13 Apr | 38.6 | 151 |
| MUE | 1MR | 1972 | 2019 | 1650 | 47 | | 20 Nov | 29 Apr | 57 | 161 |
| 7ZU | 7SC | 1951 | 2010 | 1660 | 59 | | 16 Nov | 14 Apr | 37.9 | 151 |
| CAV | 7CA | 1969 | 2020 | 1700 | 51 | no | 24 Nov | 25 Apr | 57.8 | 154 |
| BEV | 7SD | 1951 | 1982 | 1715 | 31 | | 12 Nov | 19 Apr | 45.3 | 158 |
| SAM | 7SD | 1980 | 2020 | 1715 | 40 | | 19 Nov | 17 Apr | 32.6 | 152 |
| BIV | 5BI | 1969 | 2014 | 1770 | 45 | | 11 Nov | 30 Apr | 53.3 | 166 |

**Table A 1: Station pairs, available parallel periods, mean elevation, and number of parallel years. HN indicates whether a station pair was used for HN analyses. mDstart and mDstop are mean onset and disappearance dates, mHSavg indicates the mean average snow depth, and mdHS1 the mean number of days with snow on the ground.**




| Station | Dstart_Klein | Dstart | Dstop_Klein | Dstop | Period |
|---------|--------------|--------|-------------|-------|--------|
| ANT | 2.4 | 2.5 | -5 | -5.9 | 1967-2016 |
| 2AN | | 2.5 | | -4.2 | 1967-2016 |
| BOS | 2.4 | 2.8 | -7.3 | -2.1 | 1962-2014 |
| 6BG | | 2.8 | | -4.2 | 1962-2014 |
| DAV | 3.1 | 1.6 | -4.4 | -2.5 | 1966-2006 |
| 5DF | | 3 | | -5 | 1966-2006 |
| SMM | 2.1 | -1 | -4.2 | -6.7 | 1979-2012 |
| 7ST | | -2.5 | | -5.2 | 1979-2012 |

**Table A 2: Absolute changes for Dstart and Dstop in days per decade. Dstart_Klein and Dstop_Klein are taken from Klein et al. (2016) and cover the period 1970 whereas the periods for our station (pairs) are indicated in the table.**

**Acknowledgement**

We thank Johannes Aschauer (SLF) for his valuable work in sussing out the metadata mess and contribution to the data set. We thank Francesco Issota (MeteoSwiss) for providing the sunshine duration (Sdur) data set and Wolfgang Schöner and Gernot Resch for valuable comments. This work is financed by the SNF.

**Data availability**

The basis for all our analyses, the indicator data set, is available here: https://doi.org/10.16904/envidat.218.

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
