# Peer review of "Local-scale uncertainty of seasonal mean and extreme values of in-situ snow depth and snow fall measurements"

_The Cryosphere, 2021_

## Referee Comment (RC2)

[referee-annotated manuscript omitted]

---

## Referee Comment (RC3)

Specific comments to manuscript TC-2021-125 by Buchmann et al.:

Page 1, lines 22-24: I don't understand the meaning of that sentence, consider rewriting.

Page 2, line 41: 1-km scale

Page 3, lines 78-79: Please explain. Measurement routines for new snow are not familiar for all readers and it may remain unclear, why solely taking the difference of to subsequent snow depth measurements is regarded as wrong.

Page 3, lines 70-90: Here you introduce a lot of abbreviations for your measurands and calculated variables. I am not sure, if all those abbreviations are necessary. Their unfamiliarity is severely hampering the reading flow later in the manuscript. At least, consider a table which you often refer to, to help the reader finding easily back to your definitions throughout the text or(and) simply try to use less abbreviations. I personally think that you also could just continue using "average snow depth" instead of "HSavg" during the entire text. Off course you probably still want to use abbreviations in figures with limited space.

Page 6, line 122: why do you refer to equation 2 here?

Page 6, line 123: this equation is probably not necessary. I suggest writing: …. Absolute changes defined as the difference between the fitted value at the end and the fitted value at the beginning of the time series.

Page 6, line 127: Also here I do suggest to just explain in writing, I don't see any increased value in this equation, it may actually confuse a bit (I pondered sometime about the meaning, as I could not find the time period of 60 days you describe in the sentence before).

Figure 1: Figure captions should be more stand-alone. Either refer to a table which explains the abbreviations your simply restate their meaning in the figure caption here.

Page 8-9, Section 4.2:

- I do find this entire section difficult to read because of the (already mentioned) extensive use of abbreviations. As mentioned before, add an easy to find and cite table or reconsider the use of some of the abbreviations.
- I suggest the use of "panel" or "panels" instead of "plot/s" whenever you refer figure parts
- line 184: consider writing: "…analysis due to median winter seasons shorter than 60 days" – I assume that is the threshold you have set?
- Line 192: Did you find this "empirical value" during this study or earlier? Or is it in use elsewhere, please elaborate or cite.
- Line 194: please add some concrete numbers to explain "smaller" – also some of the median differences for the ablation period are very high compared to the majority of the lines, I can't find this mentioned anywhere in the text.

Figure 2: Please use A,B,C,D in the figure caption, maybe add also top and bottom panels, left and right panels where appropriate.

Page 12, Line 211ff: Beside the mentioned exception PIO, I also see that GSS, RIE and ZER (the latter two to a smaller extend) show larger variations in spring than autumn. Also, the non-typical large difference of 18 days at 5KK remains uncommented. I'll suggest commenting those in the text.

Page 12, line 215-216: Here you write the stations are aligned from shortest to longest periods in Figure A2, while you state an alignment according to elevation in the figure caption of Figure A2, please correct

Page 12, lines 217ff: Even if belonging to the minority; I do think it is worth mentioning the non-typical behavior of BEV and MVE. Are there any reasons why those stations are so different?

Page 12, line 225: Could you describe the different definitions of snow onset and disappearance?

Page 12, line 227: I disagree that the absolute changes for especially Dstop by this study and by Klein et al (2016) are "very similar" in all cases. While some numbers are differing just by a decimal day, differ some by a factor two or in one case (BOS) even by more than a factor 3. Even if it just "days", I do think this justifies a more accurate description of the size of the differences.

Figure A2: mismatch between description of alignment in figure caption and describing text. Please correct.

Page 14, line 243: This sentence sounds strange as it reads that 64 hours amount to 1 hour a day, Please consider rewriting.

Page 14, line 248. Which abbreviation do you use for the station pair Klosters? Would you have any material to illustrate the "no difference on the first glance" and the differences you nevertheless describe? It is difficult to understand that it is not possible to see the difference between stations that close to building constructions. Please give an example on the metadata you do have and the findings you obviously have made on-site.

Page 16, line 262: I think the referral to figure 5, can already be added to first sentence in section 4.5.

Section 5, conclusions:

> In case you are keeping the abbreviations in the rest of your paper (see comments above), I would really recommend the use of complete variable descriptions here in the conclusions, to accommodate readers which start their reading with the conclusions.

> Page 17, line 275: which engineers are you referring to here?

> Page 18, lines 301-302: Is that speculation or do you have any research on that?

> Page 18, lines 304-307: while this is an important factor to consider for the analysis of past time series, it is nowadays possible to get a better set of metadata. Why don't you recommend a set of metadata which ought to be recorded for each station and what kind of locations should be avoided.

> Page 18, line 306-307: What exactly do you mean with your last sentence? Is that a wish for sites in Switzerland, worldwide? Only parallel sites? Is it enough with one of the sites in case of parallel measurements?

The numbering of the tables and figures are somewhat odd, but it seems that some of the figures/tables are supposed to be in the annex despite they appear mixed with the "normal" figures/tables of the paper. Some of the figures numbered with A* are not less used than these "normal" figures, please consider to add those figures in the main document. Also, even if not stated

on each figure/table, please insert all necessary information for understanding the figures/tables directly in the caption, independent from the describing text in the manuscript.

---

## Author Comment (AC1)

**Author responses**

**Referee 1**

We thank referee 1 for the valuable comments. Please find our detailed answers (in blue) below.

**Specific comments**

*(1) I would suggest to change the title to "Local bias of…" and to avoid the term "uncertainty" here and in the discussion. "Local bias" is used most in the manuscript, and is in terms of locally specific characteristics in snow accumulation at the station just a part of the overall uncertainty in local snow measurements in addition to e.g. measurement errors. A complete analysis of the uncertainty in terms of variability of the local bias might include not only the mean values, but also max/min/range/sigma of the seasonal values (see detailed comments/ L98). "Local variance" or "Local variation" might by another options more presenting the local effects of station setting as mentioned in the manuscript (e.g. soil, buildings, …).*

[Answer]: We agree and avoided the term "uncertainty". We changed the title to:
Local-scale variability of seasonal mean and extreme values of in-situ snow depth and snowfall measurements

And used "local bias" or variability instead of "uncertainty in the manuscript.

*(2) The presented analysis is an update of the work published by Buchmann et al. (2021) with an increased number of station pairs for a longer time period. Nevertheless, basic methods are the same, and results do not differ fundamentally. I recommend the authors to accordingly present the new findings and differences between both studies.*

[Answer]: Buchmann et al. (2021) (*https://doi.org/10.16904/envidat.218, 2021)* is the data set for this paper. We cited the data set using the doi instead of the author (year) in order to make it less confusing. Further, we rectified the erroneous  years in the references

Buchmann et al. (2021) focused on the robustness of snow climate indicators, regardless of any local influences, using station pairs with data during the same 25-year period. Our focus in this study is on local bias, introduced by different local factors. We use an extended version of the parallel data set, both in number of available stations and available years. Further, we introduce snow onset and disappearance dates, extreme value analyses, and potential sunshine durations to conduct our analyses. We kept the RPD as an error metric, to be able to compare our findings with previous results.

We slightly rephrased the introduction to clarify the differences.

Using and extending the data set of parallel time series introduced by Buchmann et al. (2021) in available number of stations, months and years, enables the investigation of the impact of the above-mentioned bias introduced by sometimes not ideal measuring locations, hereafter referred to as "local bias" or variability. Introducing snow onset and disappearance dates, as well as extreme value analyses, we strive to answer the following questions:

*(3) In the current version of the manuscript, results and discussion are presented right together. I suggest to separate both and to align the discussion linking the different results (some biases are related to the same source of error). Further, the conclusion are rather to long and can be shortened by several sentences (see specific comments) to be more condensed. The conclusion might present*

*aspects on the order of local biases, that might be negligible and what the local station bias means for the interpretation of climatologies (trends) based on single (mountain) stations.*

[Answer]: We rephrased and shortened the conclusion and added another chapter 4.6 covering metadata. We tried the suggested structure in an earlier draft but changed it to the current structure after in-house revisions as we think it is more reader friendly that way.

**Detailed Comments (L = line)**

*L10: Remove daily scale and change to: "Daily measurements of snow depth..."*

[Answer]: done

*L22: Here and throughout the manuscript: for me it was difficult to follow with the shortening with the brackets. At least it stopped the flow in reading. Please consider to present in two separate sentences.*

[Answer] : done
The highest percentage (90%) of station pairs with uncertainty less than 15% is observed for days with snow on the ground with 90%. The lowest percentage (30%) of station pairs with uncertainty less than 15% is observed for average snow depth.

*L47: Here and throughout the manuscript: there appears on redundant pair of brackets in the citation using e.g.*

[Answer]: We agree and removed the one pair of brackets.

*L48: Please add "half year" in front of period*

[Answer] :We changed period to six months.

*L56ff: the research questions are announced but not formulated as such. In addition methods are presented here (point 2). Please rephrase this part to list the analysed parameters or formulate the RQs*

[Answer]: We agree and rephrased said part.

*L72: remove the brackets and write: "...within a distance of 3 km horizontally and 150 m vertically..."*

[Answer] : done

*L72/73: 1 km: Please fix the space between number and unit such that it is not seperated at the line break (here and in the entire manuscript)*

[Answer]: done

*L74: 1770m: This might be a high elevation for station pairs, but it is not in context of mountain stations and snow measurements. Would be nice to see a discussion if and how the results are expected to be scaleable to higher located mountain stations.*

[Answer] : We agree, 1770 m a.s.l. is not a high elevation in absolute terms, however, it is for parallel measurements in Switzerland. Our focus here is solely on parallel measurements. The scaleability is limited by the fact that there are practically no stations between 1800 and 2500 m a.s.l. The few existing long-term measurements at high altitude in Switzerland are located around 2500 m a.s.l. So any scaleability attempts involve a lot of speculation and that is way we don't want to delve into that.

*L75/76: Shift the dates of the 77 year time period (1943 to 2020 ;*

[Answer]: We rephrased the sentence to:
It includes one station pair with 77 years of parallel data (1943-2020) and 10 station pairs with more than 50 years of parallel data, and incorporates a total of 1338 station-years covering the time period from 1943 to 2020.

*L76) directly to this number (L75). End the sentence after "station-pairs" in L76.*

[Answer]: see previous comment

*L86: since all measurements are made in the past, please present more detail here.*

[Answer] : We rephrased the sentence.
[..] until about two decades ago, only approximate coordinates were recorded.

*L98: since time periods are presented, the analysis of the distribution of seasonal/monthly values within the time span would be of interest, too. Please consider to give those numbers.*

[Answer] : In Table A1 and Figure A1 we show mean values of average snow depth and days with snow on the ground for each station pair as a selection of statistical properties. As there are a lot of different time series, such detailed information would in our opinion just distract from the main findings and go beyond the scope of this study. However, as we are currently working on a separate data paper focusing on the parallel data set, we think that would be a more appropriate place to go much more into detail concerning the individual time series.

*L104: To my knowledge the standard hydrological year of mid Europe is defined by 1.10. to 31.09.. Why did you use the period from 1.9. Please give at least a citation.*

[Answer] We used 1.9.-31.8. to make sure to capture entire seasons (especially important for analyses of onset and disappearance dates, as some of our stations can have snow before 1.10. We added a sentence to clarify our intentions.
To be able to capture all onset and disappearance dates, we defined the current hydrological year as the period from 1 Sep of the previous year to 31 Aug of the current year.

*L121: Consider to use an abbreviation of "local bias" for further use in the manuscript*

[Answer] : Since our manuscript has already quite a lot of abbreviations (as pointed out by referee 3), we don't consider it wise to add any more. However, as added in the introduction, we use "local bias" and variability synonymously, hence improving the reading process by not overly repeating "local bias"..

*L138: The finding that all outliers are produced by the same station pairs should be discussed in terms, if such station-pairs are representative at all.*

[Answer]: Our aim is to analyse the local-scale variability as close to reality as possible. The station-pairs and their environment are given and we don't want to focus on 'perfect' pairs only. However, the parallel analysis allows to identify 'problematic' stations that can then be avoided in further applications. We touched upon that issue in our conclusions.

*Figure 1: The RPD for all HN parameters is between 10 and 20%. Please present this in the results. It would be interesting to see the statistics on the variability of the parameters when moving a 30-year period over the entire time series of the three station pairs with most parallel years.*

[Answer]: a) We added the information about HN in the results.
For a majority of station pairs, all HN variables are below 20%.
        b) Thank you for bringing up that idea. Our RPD are mean values over the whole time series and thus do not reflect any fluctuation of the RPD with time. This is by design, as in this study we want to compare various station pairs and investigate the overall variability; and not the variability within each individual station pair. We calculated the standard deviation of RPD for the three longest station pairs for a moving 30-year window. This approach provides insight information about the variability of temporal evolution of RPD. But in order to be comparable with all the other stations in the data set (some of them way too short for this approach) we decided against using it. Moreover, as the temporal evolution of RPD is not our main focus in this study. However, this approach might be very helpful in investigating breaks in these series which is another project we are currently working on.

*L152: Replace "entire" ba "half-year"*

[Answer] : Changed to: six months

*L207: Replace "uncertainty" by "local bias"*

[Answer]: Changed to: variability

*L208: Figure A2: Please consider to put the Figure in the manuscript.*

[Answer]: Wee agree and put the figure in the manuscript. Now Figure 5 in the manuscript.

*L219: I did not get the sentence starting with "Among...". Please rephrase.*

[Answer]: We removed that paragraph as we agree with referee 2 (Craig Smith)

*Figure 4: Here and in the text: Consider to use Dend instead of Dstop.*

[Answer]: We changed Dstop to Dend

*L238-244: This is an good example to open a separate discussion*

[Answer]: We added a separate section 4.6 to discuss the metadata and an example

*Figure 5: What does the error-bars show? Please present in the caption.*

[Answer] We improved the caption.

*L259: "Analysis...." This is a result and has not to be mentioned in the conclusions again. Please revise the conclusion section and avoid to present redundant results and discussions.*

[Answer]: We tidied the conclusion.

*L264: Which estimation uncertainty is meant here?*

[Answer]: We refer to the uncertainty of the return level value, which is based on the method of estimating the parameters of a probability distribution by maximizing a likelihood function (maximum likelihood estimation).

*L291: What is the threshold to neglect local biases? This might be an interesting discussion for the discussion sector, too.*

[Answer]: We added a discussion of that point in section 4.6. And to soften the statement, we changed neglect to small, as there is no feasible threshold.
Generally, "local-bias" is often small and negligible for many applications, at least for the large majority of the stations.

---

## Author Comment (AC2)

We thank Craig Smith for his helpful comments. Please find our detailed answers (in blue) below.

*The paper provides some insight into the potential variability in local-area climate indices that users can expect due to measurement station location and local-scale variability in snow cover properties, and the potential pitfalls of extrapolating point-measurement-derived indices to the regional or landscape scale. The paper is interesting and relatively well written. I do have a couple of concerns that should be addressed before this paper can be published in TC. My major concerns are as follows:*

1. *I have concerns about the use of the term "uncertainty", largely in the title, abstract, and conclusions. I don't consider myself a metrology expert, but to me, "uncertainty" is a metric attached to a measurement to inform the user of the range of values to be expected when the measurement is made with respect to what the true value actually is. Therefore, each manual measurement presented in this paper would have an attached uncertainty, and that uncertainty would contribute to the overall uncertainty in the calculation of seasonal climate indices. However, the more appropriate terminology for what is actually being assessed here is "variability", or specifically, the impact of spatial variability on the indices. This suggested revision doesn't impact the interpretation of the results (in fact, the term "uncertainty" is really only used in the title, abstract, and conclusion, and not in the results) so updating these sections with more appropriate terminology should be a relatively easy revision.*

[Answer]: We agree and updated title, abstract, and conclusion accordingly; using variability or local bias instead of uncertainty.

2. *The paper presents some insight into the impact of local-scale variability of snow cover measurements on seasonal climate-related indices and offers some explanation as to why snow cover measurements can be quite variable in space. I believe that it is implied, but both the authors and the readers need to understand that it is highly unlikely that two measurements can adequately assess local-scale variability. I suggest that this point be clearly made (with references where appropriate) so as not to accidentally mislead the reader.*

[Answer]: We clarified that point in the conclusion by adding the following sentence:
Our term variability or "local bias" is only valid for the parallel analysis (two point measurements). But even so, the results give an indication of possible variations for various indices.

Specific comments from annotated pdf:

*Title: I'm not a very good metrologist, but in my opinion, this is not the most appropriate use of the term "uncertainty" given the context of this paper. Uncertainty is the range of values expected when you make a measurement as compared to what the "true" value actually is. You would certainly have an uncertainty associated with each manual snow depth measurement at each station, and that would contribute to the uncertainty in estimating the land scale mean snow depth (for example). However, what you are assessing is the local-scale variability, and it's impact on deriving seasonal*

*indices for the region. Having said that, the local scale variability is likely not assessable with only a pair of measurements, but that does not diminish the value of this assessment provided that both the author and the reader understand that (which means that it should be pointed out in the discussion or conclusions).*

*As an example, you state in the abstract that "there is hardly any difference between DJF and NDJFMA which show median uncertainties of less than 5% for all indicators." but it's not the uncertainty that is is less than 5%, but merely the difference in the indicators. For the most part, the only place that you talk about "uncertainty" is in the abstract and the conclusion, so this should be a relatively easy fix.*

[Answer]: Thank you for pointing that out. We agree and changed the term to variability.

*23-24: I found this sentence somewhat difficult to interpret. Maybe it's just me. For better clarity, perhaps make this two sentences:*

*The highest percentage of stations....*

*The lowest percentage of stations...*

[Answer]: true and done

*27: Can you clarify what you mean by "prevention measures"?*

*[Answer]: We added avalanche prevention measures. Could be the closure of a road, artificial triggering or evacuations.*

*36-37: this sentence structure is not quite right. I think this can be joined with the previous sentence for better structure and flow.*

*[Answer]: Rephrased to:*
All measurements of snow cover are dependent on the local characteristics of the site: i.e. exposure to wind or solar radiation, as well as nearby buildings or trees may have an impact on the measured snow quantities.

*38: "explain" is a better word*

*[Answer]: changed to recommend*

*40: typo*

*[Answer]: done*

*40: availability of observers? Not sure what you mean here*

*[Answer]: Rephrased*

*49: comma*

*[Answer]: done*

*55: the investigation of*

*[Answer]: done*

*65: their*

*[Answer]: done*

*71: You should use the entire name of the institute here, defining the acronym*

*[Answer]: done*

*74: As a geographer, I'm a little partial to maps. Could you add a map of your station pairs, perhaps colourizing the markers to indicate length of overlap?*

*[Answer]: We didn't want to overload the manuscript with figures and decided in favour of a table in the appendix (Table A1) instead of a map in the manuscript.*

*78: Could you use a couple of sentences to outline what the regular measurement procedures are? E.g., time of daily measurement, static stakes vs snow probes, snow boards, etc.*

*[Answer]: We added a sentence with references in L76-77.*
Measurements are taken every morning at 6:00 UTC at least between November and April (for details refer to Haberkorn (2019) and Buchmann et al. (2021)).

*87: perhaps "lack of awareness" would be a better choice of words*

*[Answer]: done*

*166: Is this speculation or do you have information to support this? You can soften this statement by saying "likely due to..."*

*[Answer]: softened accordingly*

*167: what exactly is "hardly any"? Rather, you should say "fewer than...".*

*[Answer]: done*

*189: Figure 2; Is there a suitable compromise such that the y-axis scales can be made the same in the left and right panels?*

*[Answer]: Unfortunately not. We tried several variations, but decided against it due to information loss.*

*191: This would be better stated as: "Snow depth (HS) and snowfall (HN) indicators are based on 30 and 24 station pairs, respectively."*

*[Answer]: done*

*198: You can drop the "respective" wording, if 20 out of 30 station pair differences are larger, then 10 are assumed to be smaller (unless some are equal, then you have to revise further).*

*[Answer]: done*

*212: "huge" is subjective. You are comparing variability in decline vs. accumulation periods, so simply saying "larger" would be more appropriate.*

*[Answer]: done*

*217-222: I'm not sure why this matters for your discussion. It would only matter if you are trying to draw conclusions about the climatological significance of the changes in Dstart and Dstop, and that is not the focus of this paper. You have noted how you define Dstart and Dstop for this paper and that's good enough. If anything, include the discussion about Klein et al when you discuss your methodology.*

*[Answer]: We agree and decided to drop the entire paragraph. We moved the references for the various definitions to our method section but kept the comparison with Klein et al. to put our values into context.*

*Methods L114ff:*

There are various definitions for snow onset (Dstart) and disappearance dates (Dend) depending on the application in hand (see e.g. Foster (1989), Kirdyanov et al. (2003), Peng et al. (2013), Stoone et al. (2002), and Klein et al. (2016)). However, as none of them suits our purpose and for sake of simplicity, we defined them as the [..]

*4.3 L234ff:*

Our values of temporal changes in Dstart and Dend correspond to values obtained by Klein et al. (2016). Although the time periods are not exactly the same, the absolute changes in Dstart and Dend are similar for the few stations analysed by both studies (see Table A2). This suggests that the absolute changes of Dstart and Dend are in general quite robust.

*229: ... in Dstop. I think since this section is only about Dstop, you should include it in the header.*

*[Answer]: done. As pointed out by referee 1, we changed Dstop to Dend.*

*238: above, you use site abbreviations in the discussion. I appreciate when the site names are used. I think this is appropriate given the limited number of times that you refer to a specific site. However, since the sites are abbreviated in the plots, you really should use both: site(abbreviation).*

*[Answer]: done*

*241: ...just 3 m away from the large turbine house at the power station.*

*[Answer]: done*

*255: I don't think this is defined anywhere*

*[Answer]: We changed it to MeteoSwiss and SLF.*

*258: IMO, this would read ok by just changing this to "variability"*

*[Answer]: done*

*263: this is a bold subjective statement, and I don't think that you have proven this at all. In fact, you showed that the return values for your station pairs are all within your prescribed confidence intervals.*

*[Answer]: We disagree, as Fig. 7 demonstrated that for HSmax one station pair and for HN3max three stations are outside each other's 95% confidence intervals. However, this sentence has been moved and rephrased as the conclusion was shortened on the request of one reviewer.*

*274: You like the word "huge" but IMO, it is a word better suited to telling fishing stories in the pub than in a scientific paper (maybe it's just me) but I would rather you use a less colloquial term.*

*[Answer]: done*

*291: better to use a few words here to indicate why the stations are "problematic".*

*e.g. The problem is that stations that exhibit "local-bias" are not easily detectable in the record without the availability of parallel measurements.*

*[Answer]: Rephrased*

*298: Shown in green are the...and shown in red are the...*

*[Answer]: done*

*Figure A2: You use red and green above to represent Dstart and Dstop and then switch to red and green here to represent each station of the pair. It's not a big issue, but can I suggest using a different colour pair here to avoid confusion?*

*[Answer]: We agree and changed the colours of Fig 5 (formerly Fig A2).*

*Table A1: These should be specified in the caption*

*[Answer]: We changed it to station pairs*

---

## Author Comment (AC3)

**Referee 3**

*[Answer]: We thank referee 3 for the valuable comments. Please find our detailed answers below.*

*Buchmann et al. are presenting an in-depth analysis of a large set of closely located station pairs with parallel snow measurement in Switzerland. Most of the sites have several decades of parallel measurements which gives valuable insight in possible differences, caused by the impact of the station environment in contrast to the climate/region the stations may represent.*

*The achieved results of the presented statistical analysis give an estimated magnitude of the added uncertainties which can be expected due to (the often unknown) variety in the less-than-ideal siting of the station. This information is valuable for any analysis of long-term snow measurement, especially in case of single located stations, as metadata describing the location for such kind of measurements are often lacking the necessary accuracy or may even be missing entirely.*

*The paper is written in an understandable language and the analysis method is described adequately. I do recommend the publication of the study, but do have some concerns and suggestions that should be addressed before publication in TC:*

1. *The authors introduce a lot of abbreviations for their measured and calculated parameters, station names, time periods (seasons) and analysis methods. While some of those are well-known and their use well-established and justified, are others solely defined in this paper. The shear amount is severely hampering the reading process, as the reader is required to either memorize the abbreviations or searching the meaning of the abbreviations somewhere in the text. This comment is of technical nature and should be easy to address. Please refer to the attached file with specific comments for possible solutions.*

[Answer]: We agree that too many abbreviations hamper the reading process. We inserted a table (Table 1) in the Data section and named the variables in full in the Abstract and Conclusion. Single months are also named in full throughout the manuscript.

*The description of the results sometimes lacks clarity and accuracy. Obvious exceptions are just very briefly and sometimes not at all mentioned, neither are possible reasons for those exceptions discussed. Please find some concrete examples in the attached file with specific comments.*

[Answer]: We rephrased several sentences and replaced vague words with more accurate terminology.
The general problem with exceptions and outliers is that they are extremely hard to explain. We tried to argue the case by inserting a new section 4.6 where we discuss an example and the limitations of the metadata in general. See our answers to the specific comments below.

2. *The authors were not able to conclude on the exact causes for the observed differences due to the lack of metadata for their sites. That fact is well described and creating awareness of this kind of problem that may likely exist for other long-term snow measurements is one of the main messages of the paper. I do think, however,*

*that it would be appropriate to recommend a list of necessary metadata to record for ongoing snow measurements and if possible, also give advice on which sites to avoid entirely.*

[Answer]:  list of necessary metadata is one thing, having (accurate) metadata at all is key. How best to describe the influence of a growing tree in the vicinity of a site? Influence of a structure (house, wall, fence et cetera) on a site? It's basically always a question of resources and will, but at least the exact coordinates should be known, ideally accompanied with pictures of the surrounding area and frequent visits to the sites.

We discussed that topic in the new section 4.6.

3. *Further, as also one of the other reviewers mentions, it is necessary to clearly state that also for the case of parallel measurements of two ideal and neighbored sites, it is likely that significant differences in the described variables may still be observed. I can imagine that some of the still existing neighbored sites (such that metadata could be achieved and described for the more recent measurements) in this study could be used to illustrate, discuss, and possibly quantify these differences and how they compare to the differences found for the long-term analysis presented in this study.*

[Answer]: We agree and created a new section 4.6, where we discuss the example of the station pair Klosters and  the use and limitations of metadata as well as possible explanations and remaining uncertainties.

*Please find more minor and specific comments and suggestions in the attached file.*

R: Specific comments to manuscript TC-2021-125by Buchmann et al.:

Page 1, lines 22-24: I don't understand the meaning of that sentence, consider rewriting.

[Answer]: We rephrased the sentence.

*Page 2, line 41: 1-km scale*

[Answer]: done

*Page 3, lines 78-79: Please explain. Measurement routines for new snow are not familiar for all readers and it may remain unclear, why solely taking the difference of to subsequent snow depth measurements is regarded as wrong.*

[Answer]: *We added a sentence with references in L76-77.*
Measurements are taken every morning at 6:00 UTC at least between November and April (for details refer to Haberkorn (2019) and Buchmann et al. (2021)).

*Page 3, lines 70-90: Here you introduce a lot of abbreviations for your measurands and calculated variables. I am not sure, if all those abbreviations are necessary. Their unfamiliarity is severely hampering the reading flow later in the manuscript. At least, consider a table which you often refer to, to help the reader finding easily back to your definitions throughout the text or(and)simply try to use less abbreviations. I personally think that you also could just continue using "average snow depth" instead of "HSavg" during the*

*entire text. Off course you probably still want to use abbreviations in figures with limited space.*

[Answer]: We agree and inserted Table 1 in the Data section. Further, we used full names throughout the Abstract and Conclusion. And single months are also written out in full. Table 1 is mentioned in the figure captions where appropriate.

*Page 6, line 122: why do you refer to equation 2 here?*

[Answer]: That was a mistake. We removed the reference and rephrased the sentence. See next comment.

*Page 6, line 123: this equation is probably not necessary. I suggest writing: .... Absolute changes defined as the difference between the fitted value at the end and the fitted value at the beginning of the time series.*

[Answer]: We rephrased the sentence accordingly.

*Page 6, line 127: Also here I do suggest to just explain in writing, I don't see any increased value in this equation, it may actually confuse a bit (I pondered sometime about the meaning, as I could not find the time period of 60 days you describe in the sentence before).*

[Answer]: We agree and removed the equation.

*Figure 1: Figure captions should be more stand-alone. Either refer to a table which explains the abbreviations your simply restate their meaning in the figure caption here.*

[Answer]: We referred to section 2 with the definitions in the figure caption.

*Page 8-9, Section 4.2:*

*•I do find this entire section difficult to read because of the (already mentioned) extensive use of abbreviations. As mentioned before, add an easy to find and cite table or reconsider the use of some of the abbreviations.*

[Answer]: We agree that too many abbreviations hamper the reading process. Instead of inserting a table which can be confusing as well, we decided to write out the variable names in full on first occurrence in each section. Single months are also named in full. We didn't want to overload the manuscript with yet another table and think this is a suitable compromise.

*•I suggest the use of "panel" or "panels" instead of "plot/s" whenever you refer figure parts*

[Answer]: done

*•line 184: consider writing: "...analysis due to median winter seasons shorter than 60 days" – I assume that is the threshold you have set?*

[Answer]: done

*•Line 192: Did you find this "empirical value" during this study or earlier? Or is it in use elsewhere, please elaborate or cite.*

[Answer]: We tried 40 days as well, but didn't see any differences. We agreed on 60 as it refers to two months.

*•Line 194: please add some concrete numbers to explain "smaller" –also some of the median differences for the ablation period are very high compared to the majority of the lines, I can't find this mentioned anywhere in the text.*

[Answer]: As the differences between the various stations are large, we only want to show that overall, they are smaller during accumulation than ablation, which can be easily confirmed by eye. We rephrased the sentence so that it now contains information about variability as well:
Here we found that the differences and variabilities observed in the first period seem to be smaller than in the last period (Figure 3).

*Figure 2: Please use A,B,C,D in the figure caption, maybe add also top and bottom panels, left and right panels where appropriate.*

[Answer]: We changed left to panels A and B and right to panels C and D.

*Page 12, Line 211ff: Beside the mentioned exception PIO, I also see that GSS, RIE and ZER (the latter two to a smaller extend) show larger variations in spring than autumn. Also, the non-typical large difference of 18 days at 5KK remains uncommented. I'll suggest commenting those in the text.*

[Answer]: The lack of reliable metadata only leaves room for speculation for ZER, GSS, and RIE; we have no explanation. Nevertheless, we completed the list of exceptions. We highlighted the issue in section 4.6 and touch upon the large difference for 5KK. But in our opinion, 5KK shows typical behaviour.

*Page 12, line 215-216: Here you write the stations are aligned from shortest to longest periods in Figure A2, while you state an alignment according to elevation in the figure caption of Figure A2, please correct*

[Answer]: rectified in text
The pairs are aligned from low (left) to high (right) elevation.

*Page 12, lines 217ff:Even if belonging to the minority; I do think it is worth mentioning the non-typical behavior of BEV and MVE. Are there any reasons why those stations are so different?*

[Answer]: In our opinion, BEV and MVE show 'typical' behaviour, see figure 4. BEV is located in the Engadin in Eastern Switzerland, whereas MVE is in the Vallais. A possible influence could be attributed to the different local climates, but would go beyond the scope of this study to verify, hence not mentioned in the manuscript.

*Page 12, line 225: Could you describe the different definitions of snow onset and disappearance?*

[Answer]: As pointed out by another referee, this whole paragraph is somewhat redundant. We decided to leave it out entirely and only mention that our values are "similar" to values

obtained by Klein et al. (2016), even though they did not analyse the same time periods and applied another definition compared to our study. We added the references for the other definitions in the method section.

*Page 12, line 227: I disagree that the absolute changes for especially Dstop by this study and by Klein et al (2016) are "very similar" in all cases. While some numbers are differing just by a decimal day, differ some by a factor two or in one case (BOS) even by more than a factor 3. Even if it just "days", I do think this justifies a more accurate description of the size of the differences.*

[Answer]: As mentioned above, we removed that paragraph in its entirety as it (1) is not strictly necessary for our analysis and (2) as you pointed out not quite accurate. Explaining the differences would go beyond the scope of this study. However, the main message is to put our values into context by comparing them to values obtained by Klein et al. (2016) for the few stations that had been analysed by both studies.

*Figure A2: mismatch between description of alignment in figure caption and describing text. Please correct.*

[Answer]: done, changed in the text to:
L225: The pairs are aligned from low (left) to high (right) elevation.

*Page 14, line 243:This sentence sounds strange as it reads that 64 hours amount to 1 hour a day, Please consider rewriting.*

[Answer]: We rephrased the sentence to:
The largest difference in a station pair (64 hours during MA) amounts to approximately one hour per day.

*Page 14, line 248. Which abbreviation do you use for the station pair Klosters? Would you have any material to illustrate the "no difference on the first glance" and the differences you nevertheless describe? It is difficult to understand that it is not possible to see the difference between stations that close to building constructions. Please give an example on the metadata you do have and the findings you obviously have made on-site.*

[Answer]: The no difference at first glance was meant to concern the metadata: same village, same elevation, but if using a proper map AND exact coordinates as well as local knowledge and metadata, differences arise.

*Page 16, line 262: I think the referral to figure 5, can already be added to first sentence in section 4.5.*

[Answer]: done

*Section 5, conclusions:*

*In case you are keeping the abbreviations in the rest of your paper (see comments above), I would really recommend the use of complete variable descriptions here in the conclusions, to accommodate readers which start their reading with the conclusions.*

[Answer]: We agree and use full variable names in the conclusion.

*Page 17, line 275: which engineers are you referring to here?*

[Answer]: civil or environmental engineers. Sentence rephrased to:
[..] which may be useful for applications where there is normally only a single time series available.

*Page 18, lines 301-302:Is that speculation or do you have any research on that?*

[Answer]: It is more an educated guess than speculation. But varying instructions combined with potentially varying executions are a likely source; but impossible to verify. We mentioned that in 4.6.

*Page 18, lines 304-307: while this is an important factor to consider for the analysis of past timeseries, it is nowadays possible to get a better set of metadata. Why don't you recommend a set of metadata which ought to be recorded for each station and what kind of locations should be avoided.*

[Answer]: WMO guidelines already give recommendations, but as we are dealing with stations that are well established, there is always balance between keeping the long-term (but maybe not perfect) measurements and begin from scratch, especially if the environment of a station is changing – which in Switzerland with its very limited space is always a danger.

*Page 18, line 306-307: What exactly do you mean with your last sentence? Is that a wish for sites in Switzerland, worldwide? Only parallel sites? Is it enough with one of the sites in case of parallel measurements?*

[Answer]: That sentence was intended as a wish for sites in Switzerland, especially in view of current efforts to optimise resources and save costs. But it obviously applies on a global scale as well, as parallel measurements are invaluable for many applications. However, as we only analysed stations in Switzerland, we added the locality and rephrased the sentence:
This means that the current number of available long-term snow measurement sites in Switzerland should at least be maintained.

*The numbering of the tables and figures are somewhat odd, but it seems that some of the figures/tables are supposed to be in the annex despite they appear mixed with the "normal" figures/tables of the paper. Some of the figures numbered with A\* are not less used than these "normal" figures, please consider to add those figures in the main document. Also, even if not stated on each figure/table, please insert all necessary information for understanding the figures/tables directly in the caption, independent from the describing text in the manuscript.*

[Answer]: We agree and improved the figure/table captions accordingly. Further, we included Figure A2 (now Figure 5) and Figure A3 (now Figure 6) in the manuscript and removed Table A2 altogether.